# Crustal Thickness of Antarctica estimated using data from gravimetric satellites.

Muriel Llubes (1), Lucia Seoane (1), Sean Bruinsma (2) and Frédérique Rémy (3)

(1) Université Paul Sabatier, OMP-GET, UM5563, CNRS/IRD/UPS, 14 Avenue Edouard Belin, 31400 Toulouse, France

(2) CNES, Department of Terrestrial and Planetary Geodesy, 18 avenue Edouard Belin, 31401 Toulouse Cedex 4, France

(3) OMP-LEGOS, UM5566, CNRS/IRD/UPS, 14 Avenue Edouard Belin, 31400 Toulouse, France

*Correspondence to*: Muriel Llubes (Muriel.Llubes@get.omp.eu)

**Abstract.** Computing a better crustal thickness model is still a necessary improvement in Antarctica. In this far continent

where almost all the rocky surface is covered by the ice sheet, seismic investigations do not reach a sufficient spatial resolution

for geological and geophysical purposes. Here, we computed a global map of Antarctic crustal thicknesses based on space

gravity observations. The DIR5 gravity field model, built from GOCE and also GRACE gravimetric data, is inverted with the

Parker–Oldenburg's iterative algorithm. The BEDMAP products are used to estimate the gravity effect of the ice and the rocky

surface. Our result is compared to crustal thickness provided by seismological studies, CRUST1.0 and AN1 models. Although

CRUST1.0 shows a very good agreement with our model, the spatial resolution is smaller with gravimetric data. Finally, we

adjust the crust/mantle density contrast considering the Moho depth from CRUST1.0 model. In East Antarctica, the density

contrast clearly shows higher values than in West Antarctica.

## 1. Introduction

The surface topography of Antarctica has already been determined precisely by the various altimetric missions ERS1/2,

EnviSat, ICESat or Cryosat-2 (Zwally et al., 2002; Rémy and Parouty, 2009; Helm et al., 2014) and more recently SARAL or

Sentinel-3 (Verron et al., 2015). Glaciologists could track temporal variations of the surface and estimate the volume changes

of the whole ice sheet (Zwally et al., 2011; Flament and Rémy, 2012; McMillan et al., 2014). Since 15 years, space gravimetry provides to the scientific community a complementary tool to observe and follow the distribution of masses inside the Earth. The main advantage is that the gravity observations have a homogeneous accuracy whatever the region of interest, i.e. over mountains, oceans or whole continents alike. Satellites are the only way to have information where gravimetric ground data

are lacking. Such observations could provide to the glaciologists the thickness of the ice cover and the temporal tracking of the ice sheet. Launched in 2002, the GRACE mission provides monthly or 10 days temporal grids (Lemoine et al., 2007; Foerste et al., 2008; Landerer and Swenson, 2012). It allows the computation of mass balance, annual or seasonal cycles (Ramillien et al., 2006; Llubes et al., 2007; Peng et al., 2016; Ramillien et al., 2006; Williams et al., 2014). In addition, temporal variations in the gravity field can be used for climate or global change purposes as a contribution to the sea level rise

equation (Jacob et al., 2012). Jointly with altimetric data, space gravimetry is used to separate snow and ice contribution (Memin et al., 2014).

Glaciological studies also need the spatial ice thickness patterns, if only to simply estimate the volume of the polar ice sheet. When modelling the ice dynamics a crucial parameter is the ice thickness and all simulations of ice thinning in response to climate forcing include this parameter as a main factor (Ritz et al., 2001). For this reason, the community made a real effort to

collect and combine all the available sources of information to compute the most complete maps over Antarctica, named BEDMAP (Lythe et al., 2001). The BEDMAP consortium provides grids of ice-surface elevation, ice thickness and bedrock subglacial topography. However, under the ice surface that covers nearly 99% of Antarctica, ice thickness is still unknown in areas without any ground data. In fact, the unobserved areas represent more than 360000 km$^2$, and very large regions without any information are in the map. These areas without observations are too large to be interpolated. Satellite gravimetry could

fill-in the missing ground data, and can be used to estimate the ice thickness (Fretwell et al., 2013).

Furthermore, as gravimetric observations are influenced by all mass distributions, they can also reveal deeper information, such as the Moho depth. Considering this purpose would be useful to geological studies, and help to understand the formation of the continent. Actually, Antarctica is still the least well-known land area. The scientific community needs improvements of the available crust models, at a continental scale but also at a more detailed scale. Several geological formations, detected in

the bedrock surface, could be explained if we know more about the crustal structure. In the past, some studies computed the

crustal thickness patterns from previous space gravimetric missions – CHAMP, launched in 2000 (Llubes et al., 2003), and GRACE (Block et al., 2009; Llubes et al., 2003). GOCE, a third satellite mission dedicated to observe to higher spatial resolution of the gravity field with high accuracy, could provide more detailed maps of the Moho limit. The other way to obtain information about this limit is to look at seismology. Some studies already used seismological data to constrain the computation

of crustal thickness from gravimetric data (O'Donnell and Nyblade, 2014). But the comparison is limited to a few locations and having a good correlation coefficient between both types of crustal thickness is not trivial. Actually, only space gravimetry can cover the Antarctica continent with a complete and dense dataset.

The aim of this paper is to provide a map of crustal thickness variations of the Antarctica continent based on the most recent combined space gravity field, named DIR5 (Bruinsma et al., 2014). We postulate that subsurface contributions can be properly

removed using BEDMAP1 grids to compute the Moho topography. Then we discuss our choices replacing the gravity field by the only GRACE and LAGEOS model EIGEN-GRGS.RL02bis (Bruinsma et al., 2010) , or preferring BEDMAP2 products. Finally, we compare our crustal thickness map to global models issued from seismological data, CRUST1.0 (Laske et al., 2013) and AN1 (An et al., 2015). The CRUST1.0 model allowed us to adjust the density of the crust in Antarctica and to map its variations.

## 2.  Datasets

### 2.1  Gravity field solutions

European Space Agency's (ESA) first Earth Explorer mission GOCE, Gravity field and steady state Ocean Circulation

Explorer (Drinkwater et al., 2003), was launched on 17 March 2009 and reentered the atmosphere on 11 November 2013. The Science Mission lasted from 1 November 2009 to 22 October 2013. GOCE was dedicated to mapping the static gravity field with a high spatial resolution of 100 km, which required measuring from a low altitude of about 250 km.

Data from GOCE (gravity gradients), LAGEOS 1/2 and GRACE were combined in the last release DIR5 (Bruinsma et al., 2014) in order to obtain a high accuracy gravity field model over the entire spectral range. This model is provided in terms of

spherical harmonics up to degree 300, but was used in this study to degree 260, i.e. 77 km of spatial resolution.

Due to the inclination of GOCE satellite, the polar regions cannot be observed. In the case of Antarctica, GOCE can only cover the continent above 83.3°S of latitude. Then, we must to notice that when we use the combined model DIR5, the signal observed below this limit latitude value is obtained by GRACE.

For our study, we computed a regular 0.25°x0.25° spatial grid of free air gravity anomalies over the Antarctica continent with the DIR5 model, which is displayed in Figure 1. The gravity anomalies have amplitudes between -120 and 111 mgal (more statistical data are provided in Table 1).

In this paper we have also used the only GRACE and LAGEOS mean gravity field model EIGEN-GRGS.RL02bis (Bruinsma et al., 2010). This gravity field model is also provided in terms of spherical harmonics up to degree 160 i.e 125 km of spatial resolution.

## 2.2 BEDMAP 1 and 2 models.

BEDMAP2 products (Fretwell et al., 2013) consist of grids describing surface elevation, ice-thickness, the seafloor and subglacial bed elevation of the Antarctic south of 60 degrees south. These products were made incorporating all available geophysical data. More details of Antarctica subglacial landscape are visible than in the previously model BEDMAP1 (Lythe et al., 2001) and the improved data coverage reveals the full scale of mountain ranges, valleys, basins and troughs. Each dataset is projected in Antarctic Polar Stereographic projection, latitude of true scale 71 degrees south, datum WGS84. All heights are in meters relative to mean sea level as defined by the EIGEN-GL04C geoid (Foerste et al., 2008). The ice thickness, bed and surface elevation grids are provided at uniform 1-km spacing.

In this analysis GOCE data is used to estimate ice thickness over the regions where observations lack. We use ice thickness given in the previous release, BEDMAP1. With the aim to represent the same range of wavelengths as the gravimetric anomalies, we filtered the difference between both ice thickness estimates to the maximum spatial resolution of GOCE, in this study 77 km. We notice that this difference is mainly at high and mean wavelengths (see Figure 2). The maximum is 1628 m, the minimum is -896 m, and the mean difference is 19 m. The new BEDMAP2 seems to improve the ice thickness all over the continent. A more complete comparison between both products is given by (Hirt, 2014).

Thanks to our procedure the Antarctica models and gravity observations remain independent. Finally, in order to have coherent

bedrock, it is calculated as the difference of the most precise topography from BEDMAP2 and ice thickness from BEDMAP1.

### 2.3 Crustal thickness models from seismology

CRUST1.0 is 1-by-1 degree global crustal model (Laske et al., 2013). This model is an upgrade of the previous model

CRUST1.0 including crustal thickness from new active seismic observations. In areas where there is a lack of seismic

observations crustal thickness is constraint by gravity observations using maps from British Antarctic Survey (Laske private

communication). Baranov and Morelli (2013) in a Moho's depth map compiling geophysical data pointed out disagreements

with CRUST1.0. The new version of this crust model may solve the problem.

Recently, another crustal model has been proposed by a Chinese team (An et al., 2015). This model is based exclusively on

seismic observations. This will provide us an independent tool for comparison with our result from gravity data, but in

Antarctica there are very few seismological observations and the Chinese model is poorly constrained. The comparison with

the CRUST1.0 model reveals large differences between them. As seen in Figure 3, from -26 to +19 km, AN1 has higher values,

mainly localized in the East Antarctic craton. West Antarctica is much thinner than the AN1 model. The latter seems also

rougher, with a less precise coast limit. CRUST1.0 has a better spatial resolution.

### 3.  Direct problem:  terrain gravity effects.

In order to estimate the gravity anomalies mainly due to crustal thickness variations, we computed by Parker method (Parker,

1973) the gravitational terrain effects using BEDMAP1 ice thickness, and sea depth and surface topography from the

BEDMAP2 model. In our case, the Parker approach is the most adequate due to the improved spatial resolution obtained using

GOCE observations. When the spatial resolution is less than 300 km we are not in the conditions to make a simple the Bouguer

approach to estimate gravity effects.  Parker proposes to use Fourier transforms to calculate the gravitational anomaly caused

by an uneven, non-uniform layer of material.

To simplify the computations, the rock equivalent topography is calculated as (Balmino et al., 2012; Hirt, 2014):



$$H_r = H_b + \frac{\rho_o}{\rho_r} H_o + \frac{\rho_i}{\rho_r} H_i \qquad\qquad (1)$$

Where $H_r$ is the rock equivalent topography, $H_b$ is the bedrock topography, $H_o$ is the ocean depth, $H_i$ ice thickness, $\rho_r$ is the rock density (2.670), $\rho_o$ is the sea water density (1.03) and $\rho_i$ is ice density (0.917).

Using Parker's method computed by Simmons (http://geoweb.princeton.edu/people/simons/software.html) as a MATLAB function, we estimate the terrain gravitational effects by considering only one interface of constant density $\rho_r$ with a topography given by eq. (1). The resulting gravitational effect filtered to GOCE spatial resolution on the pole, (i.e., around 77 km) is shown in Figure 4a.

The difference between anomalies derived from GOCE and terrain effects gives the Bouguer anomaly essentially due to crustal thickness variations. However, there are also included the errors in layer's thickness given by BEDMAP 1 and 2 models. At large spatial wavelengths, there could be geophysical signal from upper mantle that would nether be taken into account during this study. Bouguer anomalies are shown in Figure 4b.

The statistical information on GOCE free air anomalies, gravitational terrain effects and Bouguer anomalies are given in Table 1. The gravity anomalies obtained by GOCE show variations between -120 and 110 mgal. The Bouguer anomalies give variations between -250 and 360 mgal, with a very different pattern. On Figure 4b, there is a large discrepancy between the eastern part and the western part of the continent that may reflect the geological history of Antarctica. There are also a lot of details at higher resolution coming from the low degrees of spherical harmonics in GOCE gravity field. The inversion of these Bouguer anomalies will help us to better understand the crustal structure.

### 4. Inverse problem: crustal thickness estimation.

To obtain crustal thickness variations from the Bouguer anomalies inferred from GOCE observations and BEDMAP estimates (see section 3), we resolve the inverse problem using the Parker–Oldenburg iterative algorithm. To this purpose we use the 3dinver MATLAB function proposed by (Gómez-Ortiz and Agarwal, 2005). This algorithm allows computing the 3D topography of a constant density layer.

To solve the inverse problem, we must fix as input the density contrast and the mean depth of the layer. Considering that the mantle density is 3.3, we choose a density contrast of 0.63 to be coherent with our gravity terrain effects computation (rock density 2.670). According to previous studies (Block et al., 2009; Ritzwoller et al., 2001), the mean depth in West Antarctica is about 40 km and in East Antarctica is about 30 km. We fix to the mean value, 35 km, as mean depth for the whole continent for the starting computation.

A higher-cut filter is required to ensure convergence of the inverse problem (see section 2, Gómez-Ortiz and Agarwal, 2005). In our case this filter restricts frequency contents between wavelengths of 77 km and 100 km.

In order to provide information about the accuracy of the inversion, the 3Dinver program computes additionally the difference between the gravity effect of the output layer's topography and the observed Bouguer anomalies used as input. The RMS is only about 5 mgal indicating a satisfactory accuracy.

The computed crustal thickness is shown in Figure 5. Spatial variations from 29.7 to 51 km can be observed, with a mean thickness value of 41 km (see Table 1). This last is slightly stronger than the mean value of both seismological models because for these models we kept some oceanic crust in the estimation of the mean thickness. The AN1 model has a higher standard deviation compared to the others. This may come from a few geographical areas with a high noise level.

As it was already confirmed in several previous studies, Antarctica is separated into two unequal parts, testifying to the complex formation of this continent. The TransAntarctic Mountains (TAM, Figure 5) delimit the border between the thick East Antarctica (EA) and the thin West Antarctica (WA). This last one is constituted by a complicated assemblage of geological structures, still under study (Lindow et al., 2016; Heeszel et al., 2016). The Peninsula (P) and the Marie Bird Land (MBL) have the highest crustal thickness of this continental part, reaching 45 km in some places. The rest of the zone is mainly thinner than 35 km. East Antarctica's structure and history are better understood. Geological, geochemical, petrological and isotopic studies (Mikhal'sky, 2008; Dasgupta et al., 2001) suggested that the eastern part is an old craton made during several tectonic events. Successive deformation phases remodeled this crust over time. For example, in the region of Dronning Maud Land, age data reveal primary Grenvillian rocks which had been intruded by Pan-African igneous episodes (Paech, 2001). Seismic imaging contributes to understanding the evolution of this region (Kanao et al., 2011) and to mapping of variations in the

Moho depth. (Bayer et al., 2009) observed a crustal thickening from coast to inland, between 40 km and 51 km. Our gravimetric

study confirms this tendency (Figure 5) linked to the continuing thick crust in the central part of East Antarctica, whereas the

adjacent region in the west side is thinner. Also in Figure 5, the Lambert Rift (LR) clearly appears in our crustal model, and

the enigmatic structure of the Gamburtsev Mountains (GSM) seems to be independent from the surrounding geological pattern.

5    As the origin of these subglacial mountains was poorly known a specific seismic survey had been done (Wolovick et al., 2009).

Now, with our global map made from GOCE data, it is possible to follow specific geological details over the whole continent.

It can also be useful to develop and analyze an isostatic model in the interpretation of topographic structures (O'Donnell and

Nyblade, 2014).

## 5. Discussion.

### 5.1 About our choices in the computation of the crustal model

#### 5.1.1    BEDMAP1 or BEDMAP2

When computing the direct gravitational effect, we had to choose a model for all contributing layers. The best grids for the ice

thickness – and so for the bedrock elevation – are provided by BEDMAP in the most complete data compilation available. The

15    new version of these products, BEDMAP2, also includes (satellite gravity) data from GRACE and GOCE. So we preferred to

use the prior version BEDMAP1 for the ice thickness variations to have a strictly independent grid which allows us to estimate

the gravitational effect of the ice. But BEDMAP2 incorporates two orders of magnitude more data than BEDMAP1, it has a

better spatial resolution, coverage and precision at any wavelength. It is interesting to estimate the impact of BEDMAP2 in

our crustal inversion. We did the same process but using ice thickness from BEDMAP2, up to the computation of the crustal

20    thickness. The differences in the both solutions span from -3.7 to +12.2 km, but the standard deviation is only 1.2 (see Table

1 for these statistical data). The higher values are not always located in the areas completed by satellite gravity in BEDMAP2,

because the large set of new data included during the computation really improved the model. In Figure 6, the blue line shows

the areas for which no ground data available within at least 50 km. We can trust the new version of BEDMAP is closer to

reality outside the blue line limit and then our direct estimation is more accurate. Some geological details appear in the crust

25    variations estimate when computed with BEDMAP2 ice thickness. The improvement of this version is important, even when

dealing with the base of the continental crust. It would be interesting to have a version of this product that does not depend on satellite gravity data.

### 5.1.2 Profit from GOCE compared to GRACE observations

GOCE is the last of three space missions dedicated to gravity field modeling. In a lower altitude orbit equipped with a gradiometer, it measured smaller details of the gravity field. The GRACE mission was launched at a higher altitude, mainly to observe the temporal variations. But thanks to its long and still ongoing mission, it is also possible to compute an accurate geoid. The spatial resolution is less than the geoid obtained with GOCE, but we could evaluate the differences in our crust estimate from one or the other. Additionally, we hope to see crustal details in the 77 – 120 km wavelength band. It should be noted that the GOCE geoid grid is a combined solution incorporating GRACE data.

The difference between the both crustal thickness, from only GRACE model EIGEN-GRGS.RL02bis and from GOCE DIR5 gravity anomalies, clearly shows a high frequency signal, corresponding to the contribution of GOCE at small spatial scales. The discrepancies span between ±7 km, emerging from the ambient noise level. They are mainly located over the Transantarctic Mountains (TAM), the peninsula (P), and in the western part of the continent. In East Antarctica (EA), both models are really closed. The differences do not exceed 2 km, except in the Lambert Rift (LR) – see Figure 5 for location of TAM, P, EA and LR. The new GOCE DIR5 version should provide interesting details for geological interpretation and the knowledge of Antarctica structure.

### 5.2 Comparison with models using seismology observations

#### 5.2.1 The entire maps

We already compared both models issued from the seismology community (see Sect. 2.3). Clearly, CRUST1.0 looks closer to our result. We computed the difference between them and plotted the corresponding map (Figure 7). Doing the same with the

AN1 model will give differences identical to those between CRUST1.0 and AN1 because they dominate: the latter model is far from the two others.

The crustal thickness computed in this paper is on average 1.2 km higher than for the CRUST1.0 model. The larger discrepancies are located all around the Antarctica continent, because the two coast limits can differ. We also note that our crust is thinner in the TransAntarctic Mountains, in some regions of West Antarctica and in the center of East Antarctica. CRUST1.0 has smaller amplitude variations, the lowest and highest values are reached by our model (see Table 1 for statistical results).

### 5.2.2    Spatial analysis of crustal models

#### Spectral study

We computed the power spectral densities of the crustal thickness. They are shown in Figure 8.a from 0 to 1000 km for crustal thickness obtained from GOCE data, from GRACE data, for crustal thickness given by the CRUST1.0 model and by AN1.

Finally, there are very few differences at any wavelength between thickness from GOCE DIR5 model and the one from only GRACE EIGEN-GRGS.RL02bis model. This confirms the remark made in Sect. 5.1.2 about their very small spatial differences. Because GOCE DIR5 grids also incorporate GRACE data, they are naturally close to another. Except when looking at wavelengths smaller than 120 km, where the differences should be stronger. The additional improvement by GOCE data at small wavelength is not really evident.

Comparing gravity-derived and seismological models, a clear discrepancy appears between them even if the global tendency is the same. The spectral densities show more energy for crustal thickness derived from gravity observations (Figure 8.a).

In fact, we worked with the complete gravitational signal to inverse the crustal thickness. We certainly included effects from the subjacent mantle in this interpretation. These effects have a long wavelength pattern, but there is no specific signal in the spectral curve. Seismological models are not affected by these effects and they share the same behavior at short wavelength.

There is no evidence for mantellic disturbance in our result of crustal thickness. Anyway, it is not possible to isolate and correct it during the computation process.

### Profiles tracking

With the aim to compare more easily the crustal models, on Figure 8 four profiles have been extracted along a geographical path (b). They are respectively oriented S-N (c), W-E (d), NW-SE (e) and SW-NE (f). The main differences appear first over the oceans and over the two ice shelves because our inversion method is not adapted to these regions (we cut a part of the plots on Figure 8). Secondly, there are large discrepancies near the coast limit which is not the same for all models.

On the continent, CRUST1.0 is very close to our results, but it always has smaller thickness values. But as we had to choose a

mean depth to compute the Moho's spatial variations, this difference is not relevant. Looking at the four plots, we see a higher discrepancy in the North and West parts of the continent. For example, in Dronning Maud Land, CRUST1.0 gives a thinner crust than our models. Smaller scale variations are visible in the GOCE DIR5 and GRACE EIGEN-GRGS.RL02bis profiles, while CRUST1.0 is smoother.

Clearly, AN1 is very different from the three others. It shows less spatial details, with a more constant profile along west

Antarctica. In the Gamburtsev Subglacial Mountains region, the model reaches it maximum with a sudden thickening up to 65 km, over a very restrictive area (Figure 8.d). No other model shows such a local thick crust. For the moment, persons working with a seismic crustal model would prefer CRUST1.0 which is validated by gravity crustal models.

### 5.2.3    Density estimation

In our computation, we chose a 0.63 density contrast because it was coherent with previous studies and a very classical choice in such geological context. Recently, O'Donnell and Nyblade (2014) proposed a smaller contrast of 0.3 when they adjusted the crustal thickness based on gravimetric data to seismological results. Therefore, it was important to reconsider the density of the crust and the upper mantle in Antarctica. We decided to use a simple Bouguer formula to estimate the density contrast

variations, assuming that the observed Bouguer anomalies are mainly due to density variations that are not considered in the

STGE. The Bouguer anomaly map is filtered to 333 km, and we chose the CRUST1.0 model for the crustal thickness. The

result is shown in Figure 9 and it represents the variations around the crust's mean density value of 2.67 when mantle density

is fixed at 3.3. We notice that the density is slightly smaller in the western part of the continent. Few spatial variations are

visible on Figure 9. Density seems to be constant enough, from 2.72 to 2.8 in East Antarctica and from 2.62 to 2.75 in West

Antarctica. Geological studies in this last region indicate a complex assembly, made by several tectonic plates, thinner and

younger than the old eastern craton, explaining the density discrepancy. This is coherent with the results of (Tenzer and

Bagherbandi, 2013), except in Marie Bird Land where we found an  increase of density not visible in their study. However,

gravimetric observations do not need spatially varying density to be interpreted. A constant density contrast is a good first

hypothesis to compute the inversion of the Bouguer anomalies and obtain the crustal thickness.

## 6.   Conclusion.

We have provided a global map of crustal thickness, displayed in Figure 5, covering the entire Antarctic continent. Thanks to

the new GOCE gravity field model, a spatial resolution of 77 km is reached with almost the same accuracy anywhere.

Compared to previous studies, smaller details appear on the map, and the noise level is lower because we used the last available

gravity grids which compile all GRACE and GOCE data. However, the ice thickness estimation remains a key parameter

during the process. An improved ice thickness map, made from satellite observations and fully independent from gravity data,

would benefit not only glaciological but also our crustal studies.

Currently, comparing our result to seismological models shows a fairly good agreement. Some studies already tried to use

seismic observations to constrain the thickness derived from gravimetry, and seismological models begin to include

gravimetric observations. Next, a joint inversion of seismological and gravimetric data, including data from satellite missions,

will provide the next crustal model of Antarctica.

All studies concern long spatial wavelength, barely better than 100 km. Geophysical and geological studies need a higher

spatial resolution to understand the crust structure over more local regions. Antarctica is still the Earth's place where the crust



is least known. It will be a real challenge to improve the crustal model resolution even if only for specific areas. Maybe additional gravity campaigns or satellite missions will allow this progress.

Our computed crustal thickness models of Antarctica will be available on the web The International Gravimetric Bureau website (http://bgi.omp.obs-mip.fr/).

**Acknowledgements**

We want to thanks G. Laske for the information she provided to us about CRUST1.0 model. This work has been supported by University Paul Sabatier (Observatoire Midi-Pyrénées) and the French space agency CNES.

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



|  | *Minimum* | *Maximum* | *Mean* | *S. D.* |
|---|---|---|---|---|
| *GOCE Free Air Anomaly* | -120.0 | 111.7 | -4.2 | 22.5 |
| *Gravity terrain effect\** | -371.3 | 291.5 | -64.6 | 154.2 |
| *Bouguer anomaly\** | -242.8 | 373.4 | 60.4 | 149.2 |
| *CRUST MODEL* | | | | |
| *Our crust model* | 29.7 | 51.0 | 40.9 | 3.4 |
| *CRUST1.0 (Laske et al. 2013)* | 7.5 | 45.9 | 35.7 | 3.4 |
| *AN1 (An et al., 2015)* | 7.9 | 65.0 | 35.9 | 7.6 |
| *DIFFERENCE between crust models* | | | | |
| **CRUST1.0 – AN1** | -26.0 | 19.0 | -0.3 | 4.2 |
| **GOCE Bedmap1 - Bedmap2** | -3.7 | 12.2 | 1.3 | 1.2 |
| **GOCE - GRACE** | -7.6 | 7.3 | 0.0 | 0.9 |
| **GOCE - CRUST1.0** | -4.5 | 29.0 | 1.2 | 1.9 |

**Table 1: Statistical data for maps used in this study. The three first rows are in mGal, and the others are in km. The \* indicates the application of a 77km low-pass filter.**







**Figure 1. Free air anomalies in mgal from GOCE DIR5 solution.**





**Figure 2. Difference between ice thickness from BEDMAP2 and BEDMAP1 (max: 1628 m, min: -896 m, mean: 19 m). This difference is filtered to the maximum spatial resolution of GOCE (77 km) to represent the range of wavelengths that they are significant in our study.**





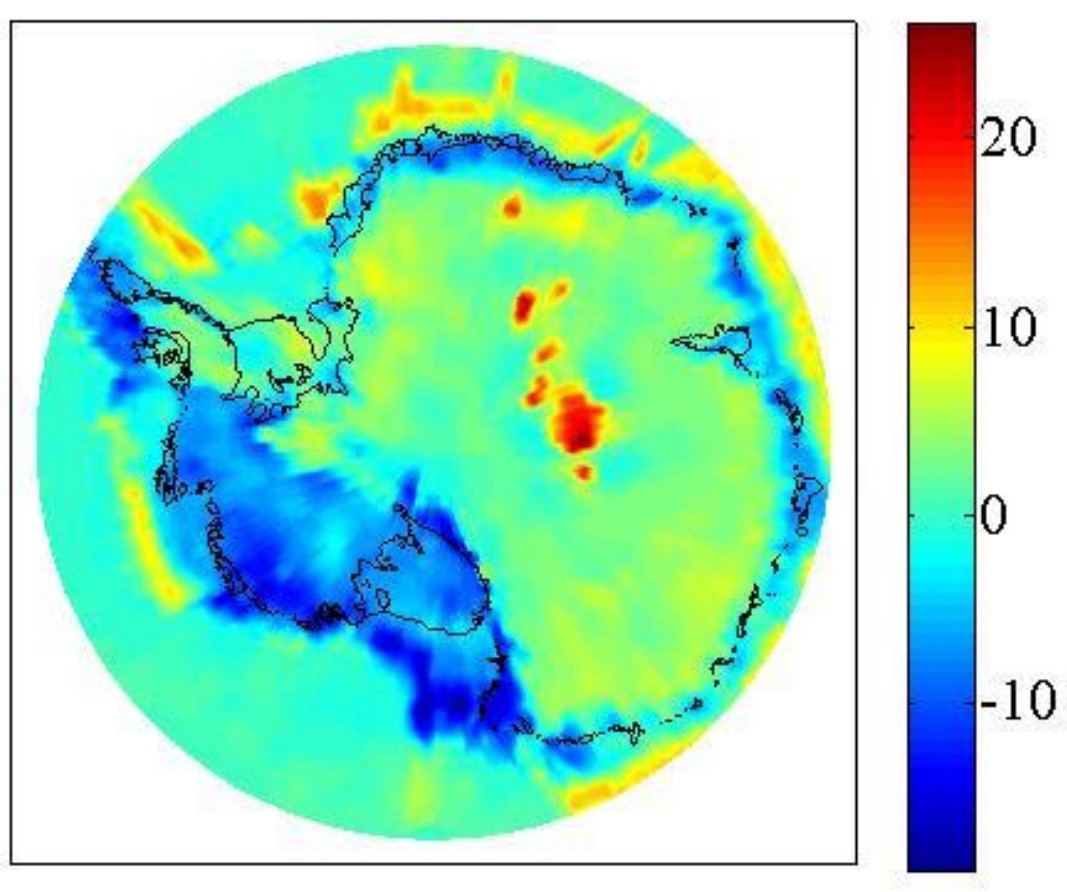

**Figure 3. Difference between crustal models AN1 and CRUST 1.0 based from seismological data. Units are km.**





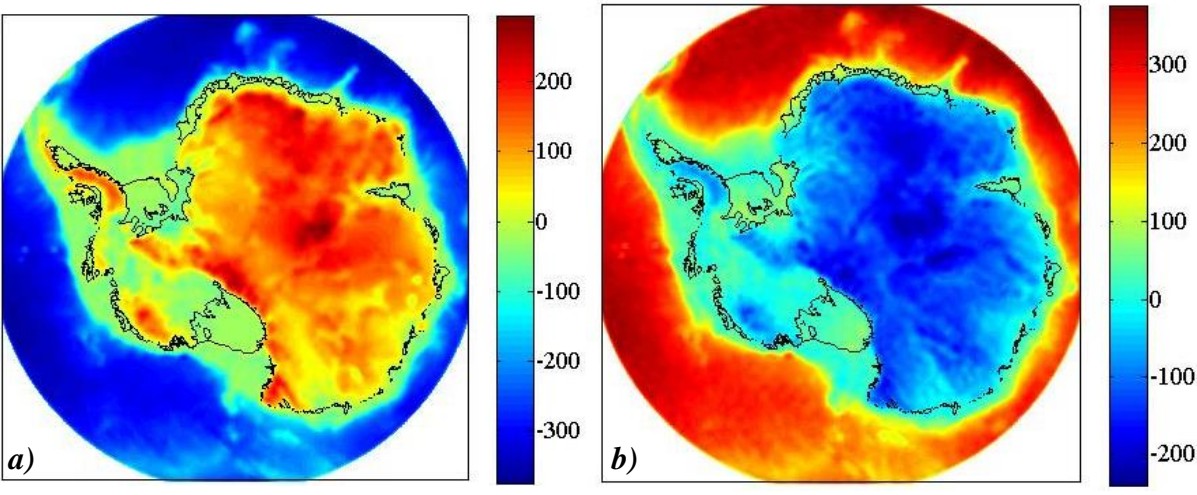

**Figure 4.a) Gravitational terrain effect derived from BEDMAP1 ice thickness, sea and bedrock topography. Low pass filter of 77 km cut-off is applied. b) Bouguer anomalies estimated as the difference of GOCE observed free air anomalies and terrain effects. (in mGals)**





**Figure 5. Estimation of crust thickness (in km) from GOCE gravity observations. Geographical locations mentioned in the text are: Dronning Maud Land (DML), Lambert Rift (LR), Gamburtsev Subglacial Mountains (GSM), TransAntarctic Mountains (TAM), Marie Bird Land (MBL) and the Peninsula (P). The large TAM separate East Antarctica (EA) from West Antarctica (WA).**





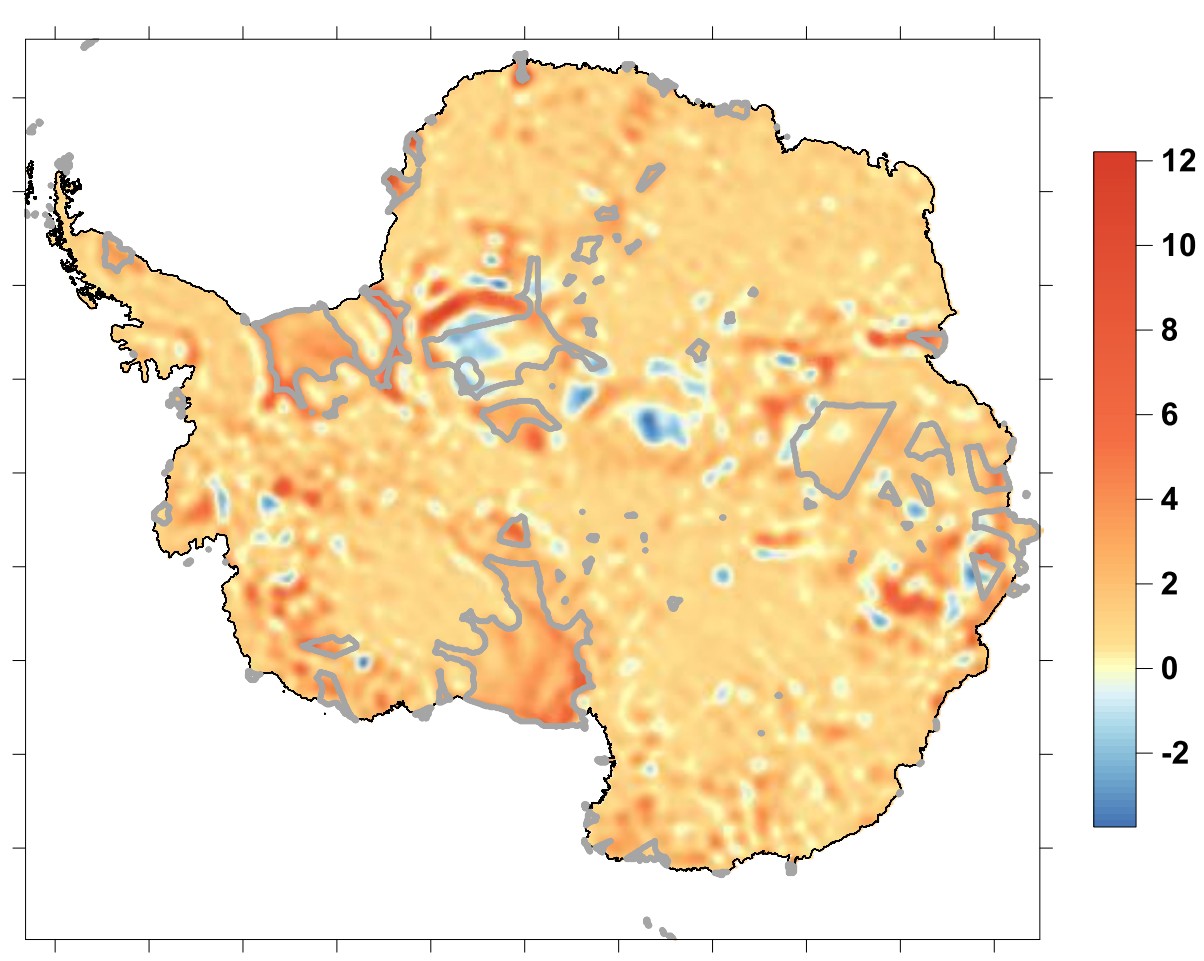

**Figure 6. Differences between estimation of crust thickness: results using BEDMAP1 minus results using BEDMAP2. The grey line delimitates the zones where GOCE observations are used in BEDMAP2. Units are km.**



**Figure 7. Difference between the crust thickness computed from GOCE and the CRUST1.0 model. Units are km.**





**Figure 8. Spectral analysis crust thickness models (a), and their thickness variations over 4 profiles directions.shown with yellow lines in fig b. The Antarctica map is converted in a cartesian X-Y coordinate system. The profiles are c) parallel to Y, d) parallel to X, e) X=Y and f) X=-Y. The south pole is located at (0,0).**





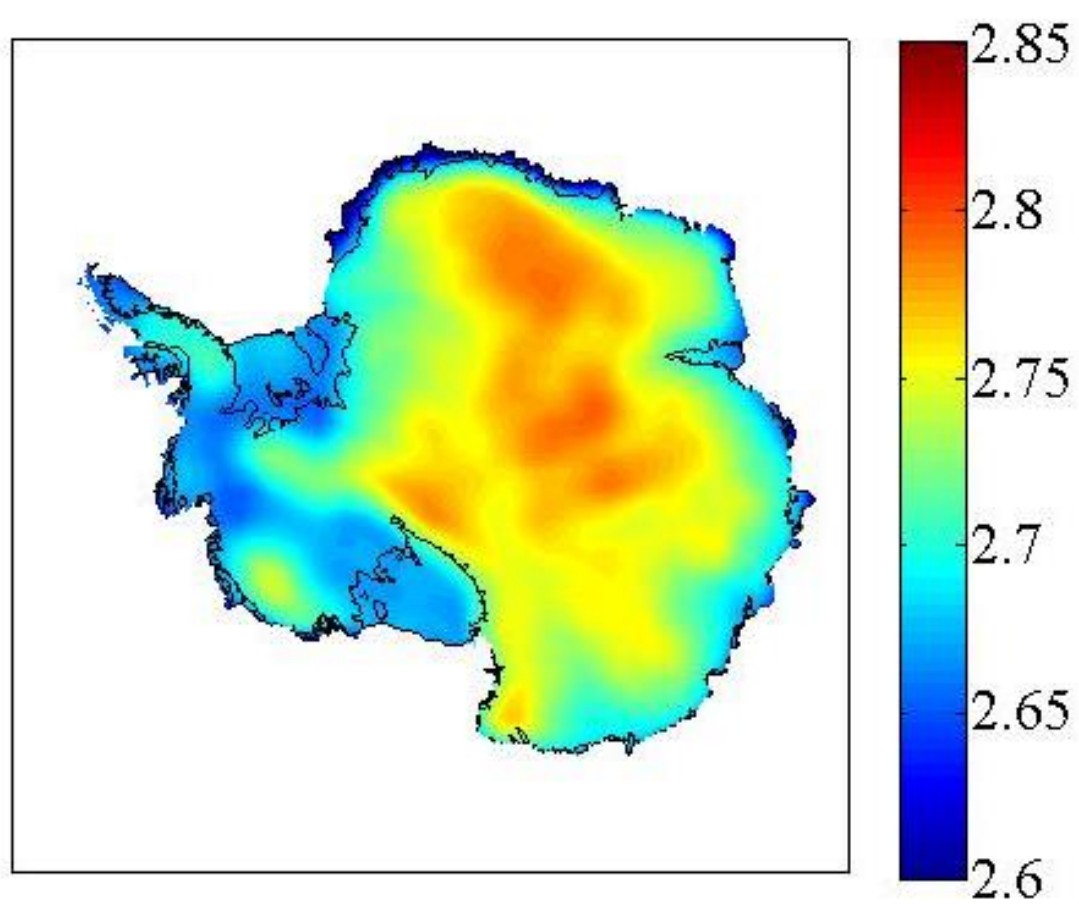

**Figure 9. Density variations estimated from CRUST 1.0 crustal thickness variations.**