# Peer review of "Crustal Thickness of Antarctica estimated using data from gravimetric satellites."

_Solid Earth, 2017_

## Referee Comment (RC1) · M. An (Referee) · 27 Oct 2017

The authors constructed maps of crustal thickness and density of Antarctica by using new data of gravimetric satellites. Since deep structure of Antarctica are least known, any effort to improve the knowledge on it is valuable. Gravity observations can offer reliable information about variations in Moho topography, but not absolute thickness values, therefore, absolute thicknesses from seismic studies are often taken as a priori constraints in gravimetric inversion. The constraints used in this manuscript are from global model, CRUST1.0, but it gives little information on Antarctic interior. Most of Antarctic interior had never been reliably measured before the Fourth International Polar Year (IPY) (2007-2008). Since 2007, the seismic works under GAMSEIS (http://epsc.wustl.edu/seismology/GAMSEIS/) and POLENET (www.polenet.org)

[Figure]

projects firstly obtained reliable information on the Antarctic interior. New results (e.g., AN1 model) showed that the crust of Antarctica is very different with those (e.g., CRUST1.0) imagined previously. However, the new results were not considered in the manuscript.

Specific comments:

(1) In section "*2.3 Crustal thickness models from seismology*", several points confused me.

The text in the manuscript about the observations used in An et al. (2015) is that "*in Antarctica there are very few seismological observations and the Chinese model is poorly constrained*" (page5, lines 11-12). This statement is completely wrong. It is true that there had ever been few seismological observations in Antarctic before 2007. However, since the IPY (2007–2008), intensive seismological surveys under GAMSEIS and POLENET projects have been conducted in Antarctica. Those observations significantly improved the coverage of seismic observations in Antarctica. As one work of the GAMSEIS project, An et al. (2015) not only used almost all seismological observations before the IPY, but also the observations of GAMSEIS and POLENET. Their model was constrained by the best data coverage on entire Antarctica to date.

After a comparison with AN1 of An et al. (2015), the manuscript concluded that "*CRUST1.0 has a better spatial resolution*" (Page 5, line 15). On the contrary, in my view, the comparison of the manuscript only demonstrated that CRUST1.0 has no valid information on Antarctic interior. Seismic studies under GAMSEIS and POLENET projects since 2007 have shown that the crust of Antarctic interior is very different from previously imagined. The results of GAMSEIS project are overviewed by An et al. (2016) (http://www.aps-polar.org/paper/2016/27/02/A160908000001). Body-wave receiver function (RF) analysis is a good method to detect Moho depth beneath seismic stations and the results are normally considered as reliable. RF studies showed that crust in central East Antarctica is thick (>50 km and even to be ˜60 km) (Hansen et al.,

2010; Feng et al., 2014), in West Antarctica is thin (˜20-30 km) (Chaput et al., 2014). However, in CRUST1.0 model, the crust in west Antarctica is >30 km thick, and in East Antarctica is <42 km (Figure 8d). The text ("*The comparison with the CRUST1.0 model reveals large differences between them. As seen in Figure 3, from -26 to +19 km, AN1 has higher values, mainly localized in the East Antarctic craton. West Antarctica is much thinner* " in " *the AN1 model*") and Figure 8d in the manuscript shows that AN1 is compatible with the RF studies. From these comparison, we can just conclude that AN1 is more reliable. It is opposite with that "*CRUST1.0 has a better spatial resolution*".

Another comparison ("*The latter seems also rougher, with a less precise coast limit. CRUST1.0 has a better spatial resolution.*") is meaningless. Spatial resolution of a model is controlled by observations, but is not related with roughness of the model. More "*precise coast limit*" in CRUST1.0 indicates that the model around coast areas of Antarctica may be constrained mostly by topography but not seismic observations. The roughness in the model of AN1 around the coast lines is related to its resolution. Figure S3 in Auxiliary material of An et al. (2015) can be taken as resolution-length map of crustal thicknesses of AN1. The figure shows that the resolution length for crustal thicknesses in AN1 is ˜120 km in Antarctica. This resolution length is similar to that (77-˜200 km) of gravity data used in this manuscript.

(2) The manuscript used the thickness of CRUST1 as constraints to analyze gravity observations. "*In areas where there is a lack of seismic observations crustal thickness is constraint by gravity observations using maps from British Antarctic Survey (Laske private communication)*" (page 5, lines 6-8). In this case, the results for those areas in the manuscript are little significant because the constraints at those areas used in this study of gravity observations are from gravity observations, and their reliability is unknown.

(3) The very-thick crust in central EANT and very-thin crust in WANT may indicate that Antarctica is special. Density may be also special in the areas. Gravity observations may be useful to detect this.

Technical corrections:

Line 10: "*another crustal model has been proposed by a Chinese team*" ==> "a regional model on Antarctic crust (AN1) (An et al., 2015) has been proposed". The AN1 model is one of results of the efforts from international (US, China, France, Japan) collaborations under GAMSEIS and partly under POLENET. The model AN1 is a regional model, but CRUST1.0 is a global model.

Lines 11-12: delete "*but in Antarctica there are very few seismological observations and the Chinese model is poorly constrained*"

Line 15: delete "*CRUST1.0 has a better spatial resolution*"

Lines 3-5: "*According to previous studies (Block et al., 2009; Ritzwoller et al., 2001), the mean depth in West Antarctica is about 40 km and in East Antarctica is about 30 km. We fix to the mean value, 35 km, as mean depth for the whole continent*" ==> "According to global 1-D model of AK135 (Kennett et al., 1995), we fix to the mean value, 35 km, as mean depth for the whole continent".

The studies (Ritzwoller et al., 2001; Block et al., 2009) did not use new seismic observation or results. It is not true that "*the mean depth in West Antarctica is about 40 km and in East Antarctica is about 30 km*". Since only a general mean value (35 km) is used, it is acceptable to cite a global 1-D model of AK135.

References

An, M., Douglas, A. W., & Zhao, Y. (2016). A frozen collision belt beneath ice: an overview of seismic studies around the Gamburtsev Subglacial Mountains, East Antarctica. Advances in Polar Science, 27(2), 78-89

An, M., Wiens, D. A., Zhao, Y., Feng, M., Nyblade, A., Kanao, M., Li, Y., Maggi, A., & Lévêque, J.-J. (2015). S-velocity model and inferred Moho topography beneath the Antarctic Plate from Rayleigh waves. J. Geophys. Res., 120(1), 359−383. https://doi.org/10.1002/2014JB011332

Block, A. E., Bell, R. E., & Studinger, M. (2009). Antarctic crustal thickness from satellite gravity: Implications for the Transantarctic and Gamburtsev Subglacial Mountains. Earth Planet. Sci. Lett., 288(1-2), 194−203. https://doi.org/10.1016/j.epsl.2009.09.022

Chaput, J., Aster, R. C., Huerta, A., Sun, X., Lloyd, A., Wiens, D., Nyblade, A., Anandakrishnan, S., Winberry, J. P., & Wilson, T. (2014). The crustal thickness of West Antarctica. J. Geophys. Res., 119(1), 378−395. https://doi.org/10.1002/2013JB010642

Feng, M., An, M., An, C., Shi, G., Zhao, Y., Li, Y., & Wiens, D. (2014). Crustal thicknesses along the traverse from Zhongshan to Dome A in east Antarctica (in Chinese with English abstract). Chinese J. Polar Research, 26(2), 177−185

Hansen, S. E., Nyblade, A. A., Heeszel, D. S., Wiens, D. A., Shore, P., & Kanao, M. (2010). Crustal structure of the Gamburtsev Mountains, East Antarctica, from S-wave receiver functions and Rayleigh wave phase velocities. Earth Planet. Sci. Lett., 300(3-4), 395−401. https://doi.org/10.1016/j.epsl.2010.10.022

Kennett, B. L. N., Engdahl, E. R., & Buland, R. (1995). Constraints on seismic velocities in the Earth from travel times. Geophys. J. Int., 122, 108−124

Ritzwoller, M. H., Shapiro, N. M., Levshin, A. L., & Leahy, G. M. (2001). Crustal and upper mantle structure beneath Antarctica and surrounding oceans. J. Geophys. Res., 106(B12), 30645−30670. https://doi.org/10.1029/2001jb000179

---

## Referee Comment (RC2) · Anonymous Referee #2 · 28 Nov 2017

This paper attempts to reconstruct the thickness of the Antarctic crust from a new satellite gravity map, which it then compares to a couple of published models. It concluded the CRUST1.0 model performs better than a recent seismic compilation (An, 2015). It has several problems, some of which stem from the poor documentation of CRUST1.0. The techniques are not original, and while the input dataset is improved on previous version, to have confidence in the work more assessment is needed (eg a comparison with receiver functions, and addressing non-isostatic support of topography).

English: In general, the English is not good, but hopefully something that could be addressed in copy editing.

Limits of Parker's method: The author's use Parker's method for deriving the terrain effect, which is appropriate for a flat plane, but may become problematic at scale where

curvature is important; this should be justified at this length scale.

Airborne gravity: The new compilation of airborne gravity by Scheinert 2016, should be at least mentioned.

CRUST1.0 circularity: The biggest issue is the comparison with CRUST1.0. The authors use old studies for mean crustal thickness constraints, which presumably was an input for CRUST1.0, admit that CRUST1.0 used airborne gravity for input, in some unspecified way (even though they reject Bedmap2 because of its gravity contamination) and come up with the same result as CRUST1.0, which is not that shocking in retrospect. In general, CRUST1.0 is not well documented enough to determine how independent of the satellite gravity data it is. There is no attempt to compare with receiver functions for Moho depth (available at An's website at http://www.seismolab.org/model/antarctica/lithosphere/ANMoho.html). Given all that, the statement that CRUST1.0 is to be preferred seems to be far too strong.

Non-Moho support: There is no attempt to address flexural or mantle support of topography, something strongly suspected for Marie Byrd Land (Chaput et al. 2014), and likely an issued for Antarctica, given the thick lithosphere and time varying ice load.

Bedmap2: An additional issue is the use of Lythe 2001 over Fretwell 2013 for the bed topography. Figure 6 demonstrated that the authors know where Bedmap2 is using GOCE data, and certainly BEDMAP was not constrained by any data in those areas anyway. There is no value in not using Bedmap2, if you know which areas do have GOCE contamination.

Technical issues: Throughout: None of the figures have spatial scale bars or graticals. Throughout: BEDMAP2 should be Bedmap2 Throughout: Strictly speaking, BEDMAP1 should be BEDMAP. Page 7, line 10: the low RMS indicates stability, not "accuracy" Page 7, Line 17: Marie *Byrd* Land

References: An, M., Wiens, D. A., Zhao, Y., Feng, M., Nyblade, A. A., Kanao, M., Li,

Y., Maggi, A., and Lévêque, J.-J.,2015, S-velocity model and inferred Moho topography beneath the Antarctic Plate from Rayleigh waves, Journal Of Geophysical Research: Solid Earth, 120, 1, 359–383, 10.1002/2014JB011332

Chaput, J., Aster, R. C., Huerta, A., Sun, X., Lloyd, A., Wiens, D., Nyblade, A., Anandakrishnan, S., Winberry, J. P., and Wilson, T.,2014, The crustal thickness of West Antarctica, Journal Of Geophysical Research, n/a–n/a, 10.1002/2013JB010642

Scheinert, M., Ferraccioli, F., Schwabe, J., Bell, R., Studinger, M., Damaske, D., Jokat, W., Aleshkova, N., Jordan, T., Leitchenkov, G., Blankenship, D. D., Damiani, T. M., Young, D., Cochran, J. R., and Richter, T. D.,2016, New Antarctic Gravity Anomaly Grid for Enhanced Geodetic and Geophysical Studies in Antarctica, Geophysical Research Letters, 43, 2, 600-610, 10.1002/2015GL067439

---

## Author Comment (AC1) · 30 Dec 2017

Response to M. An (referee)

General comment An :  Âń Gravity observations can offer reliable information about variations in Moho topography, but not absolute thickness values, therefore, absolute thicknesses from seismic studies are often taken as a priori constrains in gravimetric inversion. The constraints used in this manuscript are from global model, CRUST1.0, but it gives little information on Antarctic interior. Âż

Yes, gravity data allow us to access crustal thickness variations with respect to an average value, and not absolute thicknesses. This is our unique constraint. We chose a 35 km value based on mean values from seismological models, not only CRUST1.0,

and from literature. Using a mean value during the inversion process has no influence on the resolution of gravimetric results. Our thickness model gives information near the coastlines as well as in the Antarctic interior. Using gravity data from space allows us to benefit from a complete coverage of equal quality, and reveals details about the crustal structure anywhere on the Antarctic continent (except inside the small zone around the south pole where GOCE data are lacking).

An: "New results (e.g. AN1 model) showed that the crust of Antarctica is very different with those (e.g. CRUST1.0) imagined previously. However, the new results were not considered in the manuscript".

Our aim is to propose an upgraded model for the crust thickness variations in Antarctica, using the last and most detailed available gravimetric field. It was interesting to confront the result with other models proposed by seismological studies. CRUST1.0 is one of the most famous, but it was built considering some gravimetric data. So, we completed our comparison choosing a model fully independent from gravity data, and only based on seismological data: AN1 is the most recent one. We considered this new result in our study, especially on figure 8 of our paper where differences between all models are presented.

Specific comments Authors reply to comment (1). An: "The text in the manuscript about the observations used in An et al. (2015) is that "in Antarctica there are very few seismological observations and the Chinese model is poorly constrained" (page5, lines 11-12). This statement is completely wrong. It is true that there had ever been few seismological observations in Antarctic before 2007. However, since the IPY (2007–2008), intensive seismological surveys under GAMSEIS and POLENET projects have been conducted in Antarctica. Those observations significantly improved the coverage of seismic observations in Antarctica. As one work of the GAMSEIS project, An et al. (2015) not only used almost all seismological observations before the IPY, but also the observations of GAMSEIS and POLENET. Their model was constrained by the best data coverage on entire Antarctica to date."

We have changed the text in page 5 lines 13-14 into: "in Antarctica there are very few seismological observations compared to gravity data, for example, in areas of East Antarctica as QML (Queen Maud Land) and AGV (George V land) (see Kanao et al. 2013 figure 1 to see the station distribution or An et al. 2015 figure 4) "

An : "After a comparison with AN1 of An et al. (2015), the manuscript concluded that "CRUST1.0 has a better spatial resolution" (Page 5, line 15). On the contrary, in my view, the comparison of the manuscript only demonstrated that CRUST1.0 has no valid information on Antarctic interior. Seismic studies under GAMSEIS and POLENET projects since 2007 have shown that the crust of Antarctic interior is very different from previously imagined. The results of GAMSEIS project are overviewed by An et al. (2016) (http://www.aps-polar.org/paper/2016/27/02/A160908000001). Body-wave receiver function (RF) analysis is a good method to detect Moho depth beneath seismic stations and the results are normally considered as reliable. RF studies showed that crust in central East Antarctica is thick (>50 km and even to be ËIJ60 km) (Hansen et al., 2010; Feng et al., 2014), in West Antarctica is thin (ËIJ20-30 km) (Chaput et al., 2014). However, in CRUST1.0 model, the crust in west Antarctica is >30 km thick, and in East Antarctica is <42 km (Figure 8d). The text ("The comparison with the CRUST1.0 model reveals large differences between them. As seen in Figure 3, from -26 to +19 km, AN1 has higher values, mainly localized in the East Antarctic craton. West Antarctica is much thinner " in " the AN1 model") and Figure 8d in the manuscript shows that AN1 is compatible with the RF studies. From these comparison, we can just conclude that AN1 is more reliable. It is opposite with that "CRUST1.0 has a better spatial resolution".

The Hansen et al. (2010) study is local and focused on the GSM (Gamburtsev Subglacial Mountains). Their results shown that the mean crust thickness estimated in the stations surrounding the GSM is about 40-45 km and reach 55-58 km at the GSM. In our study we show that the crustal thickness over GSM derived from GOCE are around 40-50 km (see figure 5 of our paper) which is consistent with other gravimetric studies (Block et al. 2009, Von Freese et al. 1999, Llubes et al. 2006). However these results are different from those based on GAMSEIS data (Hansen et al. 2010, Feng et al. 2014 ) . We have changed the text "CRUST1.0 has a better spatial resolution" into: " Including gravimetric observations (land and/or satellite) as constraints of the seismological model allows to improve the spatial resolution".

An:" Another comparison ("The latter seems also rougher, with a less precise coast limit. CRUST1.0 has a better spatial resolution.") is meaningless. Spatial resolution of a model is controlled by observations, but is not related with roughness of the model. More "precise coast limit" in CRUST1.0 indicates that the model around coast areas of Antarctica may be constrained mostly by topography but not seismic observations. The roughness in the model of AN1 around the coast lines is related to its resolution. Figure S3 in Auxiliary material of An et al. (2015) can be taken as resolution-length map of crustal thicknesses of AN1. The figure shows that the resolution length for crustal thicknesses in AN1 is ËIJ120 km in Antarctica. This resolution length is similar to that (77-ËIJ200 km) of gravity data used in this manuscript."

We have changed the text "The latter seems also rougher, with a less precise coast limit." into: "The roughness in the AN1 model around the coast line is related to its resolution ". In fact, our paper shows that the crustal thickness model derived from GOCE is closer to CRUST model. But we agree with the reviewer's comment. Our purpose was just to explain models difference, but it is not the main goal of our paper. The comparison of our results with AN1 model allows us to demonstrate the interest of taking into account gravity data as constraint in crustal thickness models, because it is the most recent model derived from only seismological observations. Firstly, we can confirm/validate spatial variations over the regions where gravimetric and seismological signals are comparable, and highlight the regions where discrepancies exist between the two approaches. Secondly, the use of gravity data improves the model resolution and makes it possible to estimate density variations. We would like to emphasize that GOCE observations have a uniform resolution and accuracy over the whole area

covered by the satellite

(2) An : "The manuscript used the thickness of CRUST1 as constraints to analyze gravity observations. "In areas where there is a lack of seismic observations crustal thickness is constraint by gravity observations using maps from British Antarctic Survey (Laske private communication)" (page 5, lines 6-8). In this case, the results for those areas in the manuscript are little significant because the constraints at those areas used in this study of gravity observations are from gravity observations, and their reliability is unknown."

We have focused our approach on a regional comparison between crustal thickness models, i.e., over the entire Antarctica continent. In addition, CRUST1.0 model is not constrained by gravity observations from satellite missions. In the case of local studies, we could use complementary land gravity measurements to improve the spatial resolution but it is not the purpose of this paper.

(3) An : "The very-thick crust in central EANT and very-thin crust in WANT may indicate that Antarctica is special. Density may be also special in the areas. Gravity observations may be useful to detect this."

Yes, we agree, this is shown in figure 9 of our paper.

Biblio : We agree to add references proposed by M. An work, and we did the technical corrections. An: "Line 10: "another crustal model has been proposed by a Chinese team" ==> "a regional model on Antarctic crust (AN1) (An et al., 2015) has been proposed". The AN1 model is one of results of the efforts from international (US, China, France, Japan) collaborations under GAMSEIS and partly under POLENET. The model AN1 is a regional model, but CRUST1.0 is a global model." Ok, we changed the text

An: "Lines 11-12: delete "but in Antarctica there are very few seismological observations and the Chinese model is poorly constrained" Line 15: delete "CRUST1.0 has a better spatial resolution""

We changed the text. See comment about R2 remarks.

An : "Page 7 Lines 3-5: "According to previous studies (Block et al., 2009; Ritzwoller et al., 2001), the mean depth in West Antarctica is about 40 km and in East Antarctica is about 30 km. We fix to the mean value, 35 km, as mean depth for the whole continent" ==> "According to global 1-D model of AK135 (Kennett et al., 1995), we fix to the mean value, 35 km, as mean depth for the whole continent". The studies (Ritzwoller et al., 2001; Block et al., 2009) did not use new seismic observation or results. It is not true that "the mean depth in West Antarctica is about 40 km and in East Antarctica is about 30 km". Since only a general mean value (35 km) is used, it is acceptable to cite a global 1-D model of AK135."

We do not say 35 km is the real mean thickness of the crust in Antarctica. We only use this value to start our computation. And we choose a realistic value, in agreement with scientific literature. AK135 is an old reference (Kennett et al., 1995). We prefer to cite the reference to AN's model (2015), which is in agreement with our starting value (see table 1 of this manuscript).

Please also note the supplement to this comment:
https://www.solid-earth-discuss.net/se-2017-107/se-2017-107-AC1-supplement.pdf

---

## Author Comment (AC2) · 30 Dec 2017

RESPONSE TO R2 :

*R2 :"It has several problems, some of which stem from the poor documentation of CRUST1.0."*

5    We agree that CRUST1.0 is poorly documented. We contacted G. Laske and asked for some details. She answered "Ice thickness was constrained using maps from the British Antarctic survey", but she did not confirm whether space gravimetric data have been included or not in the model. This is the reason why we wanted to compare with a fully independent model, like AN.1. Our aim was to show that gravimetry can constrain the crustal models and help to have a better solution than models using only seismology data.

10

*R2: "To have confidence in the work more assessment is needed (e.g. a comparison with receiver functions)"*

We take into account this possibility. We propose to merge figure 6 and figure 7 into a new figure 6. Then, figure 8 becomes figure 7 and the new figure 8 shows the comparison with receiver functions for
15   the crustal thickness (we respect the limit of 10 figures max.). This comparison is a very local test. Don't forget our model is a global and a rough resolution solution.

The text of the manuscript has been changed: we added a new paragraph at the end of **5.2.2 Spatial analysis of crustal models** (page 11, line 22). This paragraph is entitled: "Comparison with seismic receiver functions »

20   Copy of the new paragraph :

**Comparison with seismic receiver functions**

To complete the confrontation with seismic crustal thicknesses, we compare our results to those from receiver functions. We use the Antarctic Moho compilation given by An (Figure 4 and Table S1 from An et al., 2015), who selected a list of stations under the evaluation of the quality of Moho depth (more details
25   and sources can be found in the publication of An). On Figure 8, we plot the differences between the crustal thickness from GOCE and the value found in the fourth column of Table S1. Roughly, we obtain the same discrepancies than those observed with the profiles on Figure 7. In East Antarctica, our model

is thinner than seismic studies (see for example Feng et al., 2014; Hansen et al., 2010). The larger disagreement is located around the Gamburtsev Subglacial Mountains region (GSM). Seismic data show a thickening up to 60-65 km in this region (see also An et al., 2016), while gravity suggest a more regular crust with thicknesses under 50 km.

5  Conversely, in West Antarctica seismic studies find between 20 and 30 km for the crust thickness which is thinner than our crust from space gravity (Figure 8). Using receiver functions from POLENET, Chaput et al. (2013) explain that this thin crust is compatible with mantle compensation, especially across MBL dome (Marie Byrd Land). During our computation at continental scale, we had to postulate the full crustal isostatic compensation of topography. In regions with mantle compensation or with density variations,
10  our results will differ from the real crustal thickness. Specific studies have to be done in such regions, based on seismic data but also on airborne (Scheinert and al., 2016) or ground gravity data, these latter having a better resolution appropriate to local studies.

15  *R2 : "And addressing non-isostatic support of topography)."*
We will reply to this comment below.

*R2:"English: in general, the English is not good"*
We already contacted the editing service.

20

*R2:"Limits of Parker's method: the authors use Parker's method for deriving the terrain effect, which is appropriate for a flat plane, but may become problematic at scale where curvature is important; this should be justified at this length scale"*

We have also estimated terrain effects using tesseroids (Uieda, L., V. Barbosa, and C. Braitenberg (2016), Tesseroids: Forward-modeling gravitational fields in spherical coordinates, GEOPHYSICS, F41-F48, doi:10.1190/geo2015-0204.1. ), a method which takes curvature into account. The differences between crustal thickness estimates from Parker or tesseroids methods are shown in the figure below. The maximum reaches 1km, which is within our error bar.

[Figure]

10  *R2:"Airborne gravity: the new compilation of airborne gravity by Scheinert 2016, should be at least mentioned"*

We add a comment in the text and mention the publication (page 12, line 10):

"Specific studies have to be done in such regions, based on seismic data but also on airborne (Scheinert and al., 2016) or ground gravity data, these latter having a better resolution appropriate to local studies."

*R2:"CRUST 1.0 circularity: the biggest issue is the comparison with CRUST1.0. The authors use old studies for mean crustal thickness constraint, which presumably was an input for CRUST1.0, admit that CRUST1.0 used airborne gravity for input, in some unspecified way (even though they reject Bedmap2 because of its gravity contamination) and come up with the same result as CRUST1.0 which is not that shocking in retrospect"*

Our mean value of 35 km is consistent with old studies cited in the text but in fact it is also consistent with AN1 mean value (see table 1 of the paper). CRUST 1.0 used airborne gravity for input in complement of seismic data. This is not a problem because we have been working with GOCE which is only satellite gravity observations. In Bedmap2, GOCE data are used in some regions of Antarctica to constrain the ice thickness, and this is a problem because we are using the same data to obtain the crust thickness.

*R2:"In general, CRUST1.0 is not well documented enough to determine how independent of the satellite gravity data it is."*

Yes we agree that CRUST 1.0 is not well documented. We have answered this question in the beginning.

*R2:"There is not attempt to compare with receiver functions for Moho depth."*

As we said we have made this comparison and it is now included in a new paragraph "Comparison with seismic receiver functions" (page 11, line 22).

*R2:"Given all that, the statement that CRUST1.0 is to be preferred seems to be far too strong."*

We can make the same analysis using CRUST 2.0. This model cannot include space data like GOCE but the resolution is worse. In the text, we do not state that CRUST1.0 is the best model, we observe that CRUST1.0 is closer to our gravity crustal model.

*R2:"Non Moho support: there is no attempt to address flexural or mantle support of topography, something strongly suspected for Marie Byrd Land, and likely an issued for Antarctica, given the thick lithosphere and time varying ice load."*

We are talking here about small wavelength features, typically smaller than 200 km (usually, it is accepted that structures larger than 200 km are isostatically compensated). This comment concerns local studies

with other sources of information. Certainly, in some regions, we could find disagreement between our gravity interpretation and local studies as Chaput et al. (2013). But our work presents results at larger wavelengths, at the scale of Antarctica. We can't resolve very local structures: GOCE resolution is very close to the 200 km limit, and we want to provide a purely gravimetric, global model. Such a model will be interesting in further studies, looking at a specific region, with complementary data at smaller resolution (for example ground or airborne gravity data, seismic data). The combination of our gravity model with local data could be used to test if the signal comes from crust thickness variations or uncompensated topography. Finally, GOCE provides static gravity field then we cannot study time varying deformations.

The text has been changed. Page 12, line 5, we added:

Conversely, in West Antarctica seismic studies find between 20 and 30 km for the crust thickness which

is thinner than our crust from space gravity (Figure 8). Using receiver functions from POLENET, Chaput

et al. (2013) explain that this thin crust is compatible with mantle compensation, specially across MBL

dome (Marie Byrd Land). During our computation at continental scale, we had to postulate the full crustal

isostatic compensation of topography. In regions with mantle compensation or with density variations,

our results will differ from the real crustal thickness. Specific studies have to be done in such regions,

based on seismic data but also on airborne (Scheinert and al., 2016) or ground gravity data, these latter

having a better resolution appropriate to local studies.

*R2:"Bedmap2: An additional issue is the use of Lythe 2001 over Fretwell 2013 for the bed topography. Figure 6 demonstrated that the authors know where Bedmap2 is using GOCE data, and certainly BEDMAP was not constrained by any data in those areas anyway. There is no value in not using Bedmap2, if you know which areas do have GOCE contamination."*

We think it is important for the reader to see easily the regions where GOCE is used into Bedmap 2 because the crustal thickness estimations in these regions are certainly affected when using Bedmap 2 in the inversion. People who are specifically working in these regions should use crustal gravity models carefully.

*Technical issues: Throughout: None of the figures have spatial bars or graticals.*

We added spatial bars to the figures.

We made all other technical changes suggested by the reviewer.

We added the 3 references mentioned by the reviewer and we cited them in the text.

---

## Author Response (AR2)

**Answers to Reviewer comments:**

*"The paper is improved from its previous version, but still has a couple of flaws: a) the English needs a thorough scrub (unfortunately I apologize that I lack the time);"*

The English have been revised.

*"b) the circularity of CRUST1.0 in comparison to the gravity, when CRUST1.0 uses gravity as an input. The authors argue that CRUST1.0 uses airborne gravity while they use satellite gravity, but the issue is not resolution: its that the assumptions about density are the same. "*

CRUST 1 use gravity observations to estimate crustal thickness only as a input where active source seismic and receiver functions studies are missing (see Laske et al. 2013). The gravity data used in CRUST1.0 from the airborne observations have not a homogenous coverage and accuracy than in the case of satellite observations of GOCE used for computing our model.

*"Given that, I think it is too strong to say that CRUST1.0 has been validated by this product (page 11, lines 19-20). I recommend removing that statement."*

We have removed the statement "We conclude that the AN1 seismic crustal model is not validated by gravity models contrary to CRUST1.0"